# Everyday norms have become more permissive over time and vary across cultures

**A list of authors and their affiliations appears at the end of the paper**

Every social situation that people encounter in their daily lives comes with a set of unwritten rules about what behavior is considered appropriate or inappropriate. These everyday norms can vary across societies: some societies may have more permissive norms in general or for certain behaviors, or for certain behaviors in specific situations. In a preregistered survey of 25,422 participants across 90 societies, we map societal differences in 150 everyday norms and show that they can be explained by how societies prioritize individualizing moral foundations such as care and liberty versus binding moral foundations such as purity. Specifically, societies with more individualistic morality tend to have more permissive norms in general (greater liberty) and especially for behaviors deemed vulgar (less purity), but they exhibit less permissive norms for behaviors perceived to have negative consequences in specific situations (greater care). By comparing our data with available data collected twenty years ago, we find a global pattern of change toward more permissive norms overall but less permissive norms for the most vulgar and inconsiderate behaviors. This study explains how social norms vary across behaviors, situations, societies, and time.

Social norms are informal, widely shared rules that govern behavior within a group or society[1]. Foundational work on social norms distinguishes them from formal laws, highlighting their role in maintaining social order through mechanisms of approval and disapproval[2,3]. While large cross-cultural surveys regularly measure norms for morally contentious issues like abortion and homosexuality[4–6], the everyday norms governing mundane behaviors in familiar situations—such as in an office or a park—have received less scientific attention. Existing research on these everyday norms shows that while individuals within a society generally agree on the appropriateness of behaviors in specific situations, these ratings can differ significantly between societies and change over time[7–10]. However, the underlying factors contributing to this variation remain poorly understood. This focus on mundane situations aligns with an emerging trend in moral psychology to move beyond classic moral dilemmas and study the more common conflicts people face in their daily lives[11]. But most of this work has focused on dilemmas within a single culture, leaving large-scale cross-cultural variation and temporal change in everyday norms under-explored—a gap the current study aims to address.

In this paper, we propose that variation in everyday norms can be understood through the lens of societal moral values. The field of moral psychology offers several important frameworks for understanding moral judgment and its cultural variation. For instance, the Theory of Dyadic Morality posits that all moral judgments are rooted in a universal template of perceived harm[12], while Morality-as-Cooperation theory suggests that morality evolved as a suite of distinct solutions to promote cooperation[13]. Another prominent framework, the Schwartz Theory of Basic Human Values, identifies ten universal values that cultures prioritize differently[14]. While these theories provide rich, detailed maps of the moral domain, it is not clear what they imply about the appropriateness of various everyday behaviors in specific situations. Our aim is to test a parsimonious model capable of explaining broad patterns of norm variation across a large and diverse set of societies and situated behaviors[15]. For this purpose, we draw on Moral Foundations Theory (MFT)[16]. MFT suggests that moral judgments are based on a set of intuitive foundations that, we will argue, are also applicable to everyday behavior.

A key distinction within MFT is between individualizing foundations (Care, Fairness, Liberty), which focus on protecting individuals, and binding foundations (Loyalty, Authority, Purity), which focus on maintaining group cohesion and social order[17,18]. Cross-cultural data suggest that societies vary along a dimension reflecting the relative priority they place on individualizing versus binding concerns[19]. We refer to this societal-level dimension as individualistic morality. We contend that this single dimension offers a powerful yet simple tool for predicting how and why everyday norms vary[15–19]. The moral foundations terminology was developed for moral judgments and is therefore not directly applicable to everyday norms. Our approach is instead to identify everyday concerns that people recognize and

✉e-mail: kimmo.eriksson@mdu.se

examine whether they can be conceived as individualizing or binding concerns. We identify three primary concerns:

- *Inconsiderateness*: This reflects whether a behavior has negative consequences for others (externalities)[20–22]. As this violates the individualizing foundation of Care, we expect this concern to be more impactful in societies with more individualistic morality.
- *Vulgarity*: This refers to behaviors that are perceived as coarse, filthy, or indecent[23]. From an MFT perspective, this concern is linked to the binding foundation of Purity[24]. Therefore, we expect it to be less impactful in societies with more individualistic morality. While we link vulgarity to the non-consequentialist concern of Purity, we acknowledge that other theories, like the Theory of Dyadic Morality, might construe such violations as a form of indirect harm (e.g., causing offense), an overlap we will return to in our discussion.
- *Lacking sense*: This reflects a concern that a given behavior has negative or no positive consequences for the actor themselves[25]. We hypothesize this concern will be more impactful in societies with more individualistic morality, not because of a direct link to a moral foundation, but through a stronger reliance on common-is-moral heuristics[26]. Prior research suggests that where individuals rely less on traditional authorities for moral guidance (a feature of individualistic morality), they are more likely to infer inappropriateness from statistical rarity or oddness[27,28]. A behavior that lacks sense is likely uncommon and thus may be judged more harshly where this intuition is stronger.

In addition to these behavior-specific concerns, the value of Liberty, the principle that people should be free to act as they see fit, is a general individualizing concern[15]. We therefore expect it to be prioritized more in societies with higher individualistic morality and, due to the general scope of this concern, lead to more permissive norms overall.

Based on this framework, we test hypotheses about how everyday norms vary across societies and change over time. Assuming the concerns a behavior elicits are largely consistent across cultures[15,19,29] (an assumption we also test), we obtain the following Hypothesis about Societal Variation in everyday norms, addressing which societies have stronger everyday norms overall, which societies have stronger everyday norms for specific behaviors, and which societies have stronger everyday norms for specific behaviors in specific situations: More individualistic morality is associated with (a) higher overall appropriateness ratings of situated behaviors (due to liberty), (b) higher appropriateness ratings of behaviors that elicit binding concerns (vulgarity) and lower appropriateness ratings of behaviors that elicit individualizing concerns (inconsiderateness), and (c) lower appropriateness ratings of behaviors that elicit concerns about inconsiderateness or lacking sense in specific situations.

We can also use the same framework to address how everyday norms change over time, a topic of increasing interest among social scientists[30–33]. Our framework predicts that everyday norms would change if the relative priority placed on individualizing versus binding concerns shifts. Several macro-level theories address such value change, proposing different drivers for this shift, such as the diffusion of global cultural scripts (World Society Theory[34]), historical ecological pressures (Pathogen Stress Theory[35,36]), or socioeconomic development (Modernization Theory[4,37,38]). While each

theory offers valuable insights, it is Modernization Theory that most directly posits a continuous and directional global trend: that economic development fosters a value shift toward greater individualism, emphasizing liberty and care while the importance of tradition and purity declines. In other words, this describes a global increase in what we term individualistic morality. This clear directional prediction allows us to translate our hypothesis on societal variation into a Hypothesis on Change in everyday norms: Change over time in global everyday norms is characterized by (a) increasing overall appropriateness ratings, (b) increasing appropriateness ratings for behaviors that elicit binding concerns and decreasing ratings for those that elicit individualizing concerns, and (c) decreasing ratings for behaviors that elicit individualizing concerns in specific situations.

To test these hypotheses, we conducted the Global Study of Everyday Norms, a large-scale survey in 90 societies measuring norms for 15 behaviors in 10 situations. We also compare our data to a study conducted twenty years prior to examine norm change in 26 overlapping societies[8]. Figure 1 illustrates the geographical scope of this study.

## Methods
The Global Study of Everyday Norms was organized through the Global Social Norms research network and was preregistered at OSF (osf.io/qz82x) on June 21, 2023, before data collection began.

### Participants
Data was collected between July 14, 2023, and May 31, 2024. To take the survey, participants needed to give their informed consent, make a commitment to give their best answers (see below), and report an age of 18 or above. For either of these reasons, 1870 potential participants who entered the survey were not allowed to take it. Another 4913 dropped out of the survey before answering any questions about social norms and therefore did not provide any data for this paper. Data was provided by a total of 25,422 participants across 90 societies (33.4% men, 57.8% women, 1.8% other, 7.1% missing data; mean age = 27.3 years, SD 11.7). The sample comprises both students (55%) and non-students (23.5%), with 21.5% missing student status information. The preregistered sample, collected by the original data collection deadline of February 28, 2024, consists of 17,288 participants in 71 societies. Participants were recruited via various methods (e.g., email, social media, survey organizations), and compensation varied by site (e.g., monetary, course credit, vouchers). All participants provided informed consent. All data collection sites and their sample characteristics are reported in Supplementary Table 1.

### Manipulated variables
Behavior and situation were manipulated within subjects.

**Behavior (15 levels).** Twelve behaviors were taken from Gelfand et al.[8] (argue, laugh, curse, kiss, cry, sing, talk, flirt, listen to headphones, read newspaper, bargain, eat). Three new behaviors were added: rest, shout in anger, and use a mobile phone.

**Situation (10 levels).** All situations were taken from Gelfand et al.[8] Situations included a funeral, in the library, at the workplace, in a job

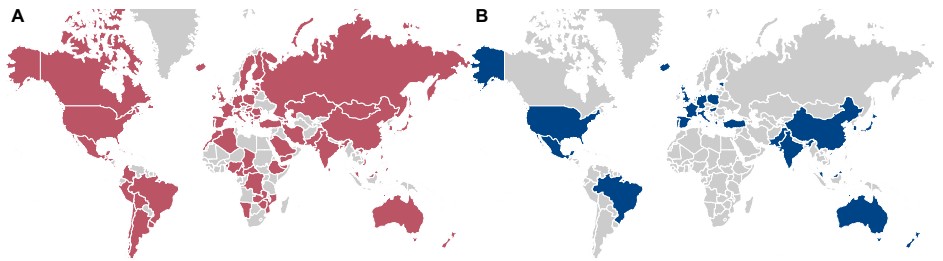

**Fig. 1 | The geographical scope of the study.**
**A** Societal variation in norms is studied across 90 societies colored red. **B** Norm change is studied in 26 societies, colored blue, for which data are available from two studies, 20 years apart.

interview, in a restaurant, in a public park, on a city sidewalk, on a bus, at the movies, and at a party.

**Excluded contexts.** In addition to these ten situational contexts, the survey included five contexts describing actor or bystander attributes (e.g., gender of actor). To maintain a clear focus on *situational* variation, which is the core of our temporal comparison with the Gelfand et al. data, and to keep the scope of this report manageable, the analysis of these non-situational contexts is beyond the scope of this paper and will be addressed in a subsequent report.

### Measured variables
**Appropriateness ratings.** The appropriateness of each behavior in each context was rated on a 6-point scale from extremely inappropriate (1) to extremely appropriate (6) following Gelfand et al. [6]. To reduce fatigue, each participant rated a random subset of combinations, resulting in an average of 5560 ratings for any given situated behavior globally.

**Concern and Commonness ratings.** To measure the concerns elicited by a behavior, participants were asked to identify the main problem for someone who disapproves of it, with options for vulgar, inconsiderate, lacks sense, no one would disapprove, or other. The commonness of each situated behavior was rated on a 6-step scale from extremely uncommon to extremely common. These tasks were also distributed across random subsets of participants, resulting in an average of 326 ratings of concerns and 377 ratings of commonness for any given situated behavior. The design choice to have fewer participants rate concerns and commonness compared to appropriateness was made to minimize participant fatigue while still gathering robust data on our primary dependent variable (appropriateness) and the characteristics of the stimuli (concerns).

**Individualistic morality (Ind-Bind Scale).** Our primary measure of individualistic morality is the 9-item Ind-Bind scale developed for this study. It measures the relative importance of individualizing versus binding concerns through dilemmas. A typical item reads: "What do you think is the right thing to do in a situation when someone must either (A) cause pain to someone or (B) be disloyal to their kin?" (the full Ind-Bind Scale is presented in Supplementary Table 2). Responses were on a 5-point scale (Definitely A, Probably A, Can't say, Probably B, or Definitely B), coded such that higher values represent a greater prioritization of individualizing concerns over binding concerns. This direct comparison format is designed to measure the relative prioritization of foundations when they conflict, a common method for assessing value hierarchies[39,40]. The scale's validity is strongly supported by its high correlation with several other related value measures collected in the study (see below and Results).

**Related value measures.** For validation of the scores on society's individualistic morality obtained using the Ind-Bind scale, several related measures were also included and aggregated at the level of societies: The difference in relevance between individualizing and binding moral foundations was measured by a version of the Moral Foundations Questionnaire[41]. This version included 8 items measuring the relevance of individualizing foundations and 9 items measuring the relevance of binding foundations (see Supplementary Table 3), providing an indirect comparison between individualizing and binding foundations. We then subtracted the average response to binding items from the average response to the individualizing items.

Freedom-of-choice values were measured by three items (acceptance of homosexuality, abortion, and divorce) phrased as "For each of the following actions, please indicate whether or not you think it is wrong" with a five-point scale coded from 1 to 5: Always wrong, Mostly wrong, Sometimes wrong, Rarely wrong, Not wrong at all. This measure is based on the Choice index in the World Values Survey[4].

Gender egalitarian values were measured using three items: "When jobs are scarce, men have more right to a job than women; On the whole, men make better political leaders than women do; A university education is more important for a boy than for a girl" with a four-point scale ranging from Strongly disagree (4) to Strongly agree (1). This measure is based on the Equality index in the World Values Survey[6].

Religious belief was measured using the item "How strongly do you believe in God (or gods)?" with a response scale from 0 to 100.

**Demographics.** The survey included items about the participant's gender, age, urban/rural background, the single-item MacArthur Scale of Subjective Social Status, and level of education (for non-students). The study additionally included items not used in the present paper (the general appropriateness of different forms of norm enforcement and perceptions of the tightness-looseness of society); no hypothesis or analysis involving these measures was mentioned in the preregistration.

### Procedure
The data was collected anonymously online using Qualtrics, with exception for (part of) the data from Mauritius and Benin where the same questions were asked face-to-face by an interviewer who recorded the responses in Qualtrics. To take the survey, participants needed to give their informed consent, make a commitment to give their best answers (see below), and report an age of 18 or above.

To pass the attention check, participants needed to respond correctly to the following survey item: "Many things may affect our judgment of whether a behavior is appropriate or not. We are testing whether or not people read questions. To show us you've read this far, please answer both 'very interested' and 'extremely interested.'" (There were five checkboxes labeled from 'not at all interested' to 'extremely interested'. As an additional attention check, we used the commitment pledge[42], which read "We care about the quality of our survey data. To get the most accurate measures of your opinions, it is important that you thoughtfully provide your best answers to each question in this survey. Do you commit to thoughtfully provide your best answers to the questions in this survey?" Respondents who selected the response option "I will not provide my best answers" did not get to proceed with the survey. Those who selected "I will provide my best answers" (96.2%) or "I can't promise either way" (3.8%) were allowed to participate, but those who selected the latter option were not counted as having passed the attention check (see next section).

### Samples used in the analysis
The analysis is performed on three samples: Preregistered, All Data, and Attention Check. The Preregistered sample ($N = 17,288$) includes data collected until February 28, except for a few societies that did not include certain questions. The All Data sample ($N = 25,422$) also includes these societies and data collected after February 28. The Attention Check sample ($N = 15,599$) excludes participants who did not pass the attention check. As shown in Supplementary Table 1, the attention check pass rate varies dramatically between societies. Given this large variance, presenting the results for the attention-checked sample demonstrates that our findings are not driven by societies with lower data quality.

### Analysis
All analyses were performed using R version 4.3.1.

**Imputation of missing data.** In the full sample, the percentage of missing values in one of the items used for the Ind-Bind scale was 4.9% including all participants from Kuwait and Saudi Arabia, where the items about violating a religious rule and breaking with a strong societal tradition were omitted. We imputed those using the mice[43] and miceadds[44] packages in R and the information from items on gender egalitarian values, freedom-of-choice values, and moral judgments. We also imputed 7.1% missing values for gender (those were mostly drop-outs who did not reach the last page of the survey with demographic variables).

**Ind-bind factor scores and measurement invariance**. As pre-registered, we performed the following analysis of the measurement invariance of the Ind-Bind scale. The analysis uses multilevel structural equation modeling (MLSEM) as implemented in the lavaan package in R[45] to test for measurement invariance of the scales across societies and estimate factor scores while accounting for potential bias due to non-invariant indicators[46]. The MLSEM approach for establishing measurement invariance requires that at least configural invariance holds. We fitted a model with all 9 indicators loaded on a single factor and allowed the residuals from the same moral foundation to covary (for example, care versus authority, care versus loyalty, and care versus purity). The two-level single-factor model with non-zero residual variance for all items at the society level (indicating there is a bias at the items) and the covariance between the residual variance from the same moral foundations (indicating that the bias is correlated) showed an acceptable fit according to the criteria CFI = 0.995 > 0.9, RMSEA = 0.015 < 0.08, and SRMR at the individual level = 0.007 and at the society level = 0.061, both <0.08[47]. Factor scores are used in all analyses. Supplementary Table 4 for factor scores for all societies, Supplementary Table 5 for factor loadings, etc., and Supplementary Table 6 for tests for cross-level/strong invariance.

**Validation of value differences**. We used the Pearson correlation to examine how the societal variation in Ind-Bind scores matches the variation in each of the related value measures included in the study, as well as how the societal variation in freedom-of-choice and gender egalitarian values obtained in our study matches the variation in the corresponding measures in the IVS, which combines the World Values Survey and the European values studies[48,49]. We calculated the mean response for the three corresponding items and averaged them over the last available wave for each country.

**Everyday concerns**. For a given behavior $b$ in a given situation $x$, we obtain measures of the concerns about the situated behavior $xb$ being inconsiderate ($IC_{xb}$), vulgar ($BC_{xb}$), and lacking sense ($NC_{xb}$) by calculating the proportion of all respondents who chose the corresponding option. The situational dependence of each concern was estimated as one minus the proportion of variance explained by behaviors according to a two-way ANOVA (behaviors × situations) of 150 situated behaviors.

After performing a median split of societies on their Ind-Bind score, we recalculated concern scores separately in each half (i.e., the proportion of all respondents in societies with high/low Ind-Bind scores who chose a certain option) and used the Pearson correlation to establish that concerns vary across situated behaviors in similar ways in low and high individualistic societies.

**Everyday norms**. $A_{xbic}$ is the appropriateness rating of behavior $b$ in situation $x$ given by individual $i$ in society $c$, recoded to range from −2.5 to 2.5, with higher values representing higher appropriateness, that is, a more permissive norm. By averaging ratings across individuals in a society, we obtain society ratings of the norm for each situated behavior ($A_{xbc}$), shown in Fig. 2.

By averaging $A_{xbc}$ across all societies, we obtain the global norm strength for each situated behavior ($A_{xb}$) shown in Fig. 3A. By averaging $A_{xb}$ across situations and centering the result on the mean across behaviors, we obtain the behavior-specific ratings shown in Fig. 3B. By centering $A_{xb}$ on the mean across situations for each behavior, we obtain the situation-specific ratings shown in Fig. 3C.

**Preregistered analysis of societal variation in everyday norms**. The preregistered analysis specified one mixed-level model to analyze the overall and behavior-specific parts of the hypothesis and a second mixed-level model to analyze the situation-specific parts of the hypothesis, but as these parts are independent of each other, we can equivalently merge the

two models into one. The analysis uses behavior-specific concerns ($IC_b$ and $BC_b$) obtained by averaging the concerns for situated behaviors across situations and centering the resulting measures at the mean across behaviors, and situation-specific concern measures $IC_{xb}$ and $NC_{xb}$ centered at the mean across situations for each behavior. The full model can be written as follows:

$$A_{xbic} = \beta_0 + \beta_1 IM_c + \beta_2 IM_c \times IC_b + \beta_3 IM_c \times BC_b + \beta_4 IM_c \\ \times IC_{xb} + \beta_5 IM_c \times NC_{xb} + \beta_6 IC_b + \beta_7 BC_b + \beta_8 IC_{xb} \\ + \beta_9 NC_{xb} + \beta_{10} IM_{ic} + \beta_{11} IM_{ic} \times IC_b + \beta_{12} IM_{ic} \times BC_b \quad (1) \\ + \beta_{13} IM_{ic} \times IC_{xb} + \beta_{14} IM_{ic} \times NC_{xb} + \beta_{15} Gender_{ic} \\ + \beta_{16} Age_{ic} + u_{1xb} + u_{2b} + u_{3c} + u_{4ic} + e_{xbic}$$

In this analysis, $A_{xbic}$ is the appropriateness rating of situated behavior $xb$ made by participant $i$ in country $c$, $IM_c$ and $IM_{ic}$ refer to the factor scores of the Ind-Bind scale for the society and the individual, respectively. $Gender_{ic}$ was dummy coded 1 for woman, 0 for man/other, and $Age_{ic}$, in tens of years, was centered on the global mean. (There were only 2% who identified as "other"; although not shown in the paper, the gender effect is not substantively different if "other" is categorized together with women instead of men.) Random intercepts were included at the levels of situated behaviors ($u_{1xb}$), basic behaviors ($u_{2b}$), societies ($u_{3c}$), and individuals ($u_{4ic}$). 95% confidence intervals were computed using the Wald method with residual degrees of freedom approximated using the Satterthwaite method. The assumptions of the model—that residuals are independent, normally distributed and unrelated to predictors and between different levels (homoscedasticity)—were generally met. Formal assumption tests were not conducted because, with our large sample, such tests are overly sensitive and of limited diagnostic value. Instead, diagnostic plots (e.g., Q–Q plots and residuals-versus-fitted) indicated that residuals and random effects were approximately normally distributed and showed no clear patterns or heteroscedasticity. Residual plots showed discrete banding consistent with the ordinal nature of the response variable, which is expected; standard inference for linear mixed-effects models with a large sample size is generally robust to this feature.

**Exploratory analyses of societal variation in everyday norms**. GDP per capita data for 2023, adjusted for purchasing power parity, were downloaded from the International Monetary Fund API. For the Caribbean society, which includes data from several Caribbean countries, we used the data for Trinidad and Tobago. The data for Sri Lanka was for 2022. For Cuba, Gibraltar, Kosovo, and Martinique, data were not available. We extend the model in Eq. (1) by including terms for GDP and its interactions with concerns ($IC_b$, $BC_b$, $IC_{xb}$, $NC_{xb}$).

To control for non-independence of societies, we use the approach suggested by Claessens et al.[50]: allowing society-level random intercepts to covary according to geographic and linguistic proximity. Bayesian models were fitted with the brms R package[51] in stan[52], using weakly informative priors that, when applied to the baseline model, yield estimates for fixed and random effects that are very close to those obtained in the main analysis. For the Caribbean society, which includes data from several Caribbean countries, we used proximity measures for Trinidad and Tobago. Proximity data were not available for Kosovo.

Perceived commonness of a situated behavior was measured by the average response to that item among all participants. The model was extended to include the main effects of behavior-specific and situation-specific commonness (calculated similarly to the behavior-specific and situation-specific measures of concerns) and their interactions with individualistic morality.

**Global change in everyday norms over 20 years**. To study norm change, we use appropriateness ratings for twelve behaviors in ten situations, which were included both in our study and the older study of Gelfand and colleagues[8]. The behaviors were: argue, laugh out loud,

**Fig. 2 | Measures of everyday norms.** The norm for a given situated behavior in a given society was measured by ratings on a six-step scale from extremely inappropriate, coded −2.5, to extremely appropriate, coded 2.5, and averaged across participants (*n* = 54.3 on average, ranging from 1 to 217 across cells). A lighter color means a higher inappropriateness rating. White cells represent missing data due to some countries not allowing some questions.

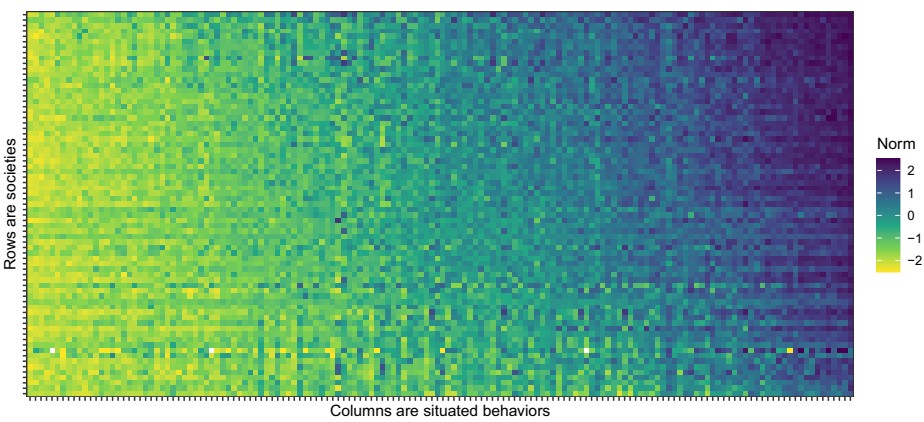

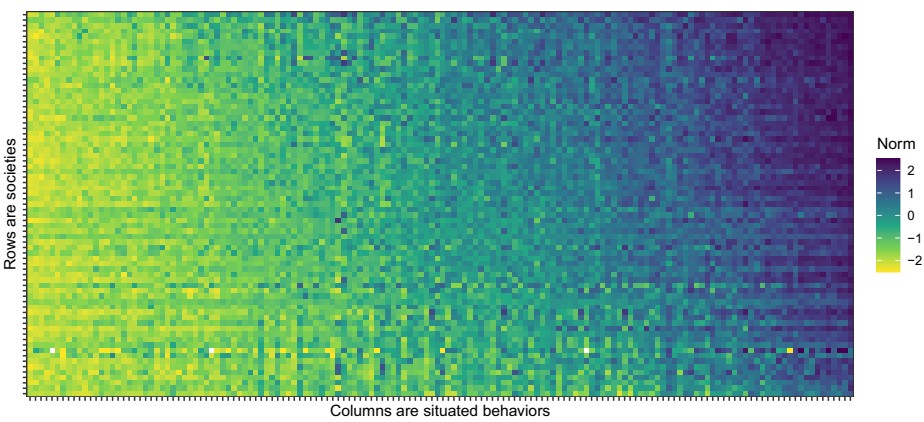

**Fig. 3 | Behavior-specific and situation-specific global everyday norms. A** A color-coded matrix illustrating the global appropriateness ratings (averaged over 71 societies) of fifteen behaviors in each of ten situations. **B** Scatter plot illustrating appropriateness ratings of fifteen specific behaviors aggregated across various situations (centered on the mean across behaviors) and their strong negative association with the sum of behavior-specific concerns about vulgarity and inconsiderateness. The index '*b*' indicates that measures refer to behaviors. **C** Scatter plot

illustrating appropriateness ratings of 15 behaviors in ten specific situations (*n* = 150 situated behaviors, centered on the mean across situations for each behavior) and their strong negative association with the sum of situation-specific concerns about inconsiderateness and lacking sense. The index '*xb*' indicates that measures refer to situated behaviors. Gray shading indicates 95% confidence intervals.

**Table 1 | Change in behavior-specific appropriateness ratings of 12 behaviors in 22 societies**

| Behavior | Mean (SD) change in rating | % of countries where ratings have increased |
|---|---|---|
| Flirt | −0.34 (0.32) | 18.2 |
| Argue | −0.31 (0.33) | 13.6 |
| Laugh | −0.19 (0.39) | 40.9 |
| Kiss | −0.07 (0.37) | 36.4 |
| Talk | −0.02 (0.31) | 45.5 |
| Sing | −0.00 (0.19) | 40.9 |
| Bargain | −0.00 (0.36) | 40.9 |
| Curse | 0.09 (0.34) | 59.1 |
| Read newspaper | 0.20 (0.19) | 90.9 |
| Eat | 0.28 (0.23) | 86.4 |
| Listen | 0.48 (0.21) | 95.5 |
| Cry | 0.56 (0.36) | 100.0 |

Behaviors are sorted by the mean change in ratings. SD is the standard deviation across societies.

curse/swear (use foul language), kiss (on the mouth), cry (shed tears), sing, talk (have a conversation), flirt, listen to music on headphones, read a newspaper, bargain (exchange goods, services, or privileges), and eat. The situations were: at a funeral ceremony, in the library, at the workplace, in a job interview, in a restaurant, in a public park, on a city sidewalk, on a bus, at the movies, at a party.

In the older study, sample sizes per society ranged from 111 to 312. All participants rated every situated behavior. Ratings were aggregated per society. To obtain global ratings we averaged ratings across societies. Behavior-specific ratings were obtained by aggregation ratings across situations for each behavior. Change scores for behavior-specific ratings (Table 1) were obtained by subtracting the rating in older study from the rating in the new study.

**Preregistered analysis of global change in everyday norms.** Mirroring the analysis for societal variation, we merge the two preregistered models to obtain a single model covering overall, behavior-specific, and situation-specific change in global everyday norms. With $t$ for time (1 for new study, 0 for the old one):

$$
\begin{aligned}
A_{xbic} =& \beta_0 + \beta_{1t} + \beta_{2t} \times IC_b + \beta_3 t \times BC_b + \beta_4 t \times IC_{xb} \\
& + \beta_6 IC_b + \beta_7 BC_b + \beta_8 IC_{xb} + \beta_{10} Gender_{ic} \\
& + \beta_{11} Age_{ic} + u_{1xb} + u_{2b} + u_{3c} + u_{4ic} + e_{xbic}
\end{aligned}
\quad (2)
$$

The model assumptions were assessed using the same diagnostics as in the societal-variation analysis and were generally met.

**Exploratory analysis of global change in everyday norms.** The analysis in Eq. (2) was also performed separately in each society (Fig. 6B).

**Summary of deviations from the preregistered analyses.** The preregistration outlined five hypotheses, numbered H1–5. The preregistered analyses for these hypotheses are reported in Supplementary Table 7. In this paper, we present two hypotheses, which together cover four preregistered hypotheses. The Hypothesis on Societal Variation contains both H1 and H3, and the model used to test this hypothesis combines the preregistered models for H1 and H3. Our Hypothesis on Change similarly contains both H4 and H5, and the model used to test it combines the preregistered models for H4 and H5. As the original wording of the hypotheses was somewhat cumbersome, we have streamlined the language. This should make it easier for readers to see that the hypotheses

match the analyses. In this paper, we do not delve into the non-situational contexts covered by preregistered hypothesis H2, as they are outside the scope of our focus on norms for situated behaviors.

### Inclusion and ethics

Local researchers were involved throughout the research process. In April 2023, members of the Global Social Norms network were asked to collaborate on this study by collecting data from their societies and coauthoring the primary publication of the study; collaborators would also be able to use the data they collect for their own publications. To ensure the local relevance of the study, collaborators were specifically asked to report whether any of the behaviors or situations used in the study would not be relevant in their society. All collaborators were also given opportunities to review and make suggestions on the entire survey and the preregistration of the study. Some funding for collaborators in low- and middle-income countries was available on request.

Local ethics review committees reviewed the study wherever this was required. To comply, a few collaborators had to exclude certain items from the survey; specifically, in Saudi Arabia, the survey did not include items on kissing, and in Kuwait, the survey did not include items on flirting, kissing, violating a religious rule, and breaking with a strong societal tradition. No personal risk for participants or researchers was expected in this study. No local research relevant to cite in this study was identified. Ethics committees and institutional review boards that approved the study protocol (or decided that the study was exempt from ethical approval) include: the Swedish Ethical Review Authority (Sweden); Macquarie University (Australia); Queen's University General Research Ethics Board (GREB) (Canada), Department of Psychology Board of Ethics, Faculty of Humanities and Social Sciences, University of Zagreb (Croatia); Université Protestante au Congo (Democratic Republic of Congo); Paris School of Economics (France); Research and Research Degrees Committee, University of Gibraltar (Gibraltar); Research Ethics Committee (REC) (Greece); United Psychological Research Ethics Committee (Hungary); Monk Prayogshala Institutional Review Board (India); Aoyama Gakuin University Research Ethics Committee (Japan); Ethics Committee of Graduate School of Informatics (Japan); Institutional Review Board Committee, American University of Kuwait (Kuwait); Sunway University Research Ethics Committee (Malaysia); Faculty (FEMA) Research Ethics Committee (Malta); University of Otago Ethics Committee (New Zealand); Ethics Subcommittee of the Macedonian Academy of Sciences and Arts (MASA) (North Macedonia); Ethical Review Board at SWPS University Faculty of Psychology in Warsaw (Poland); Scientific committee of the Center for Social Diagnosis (CDS) (Romania); Ethics Committee of the Department of Psychology, Faculty of Philosophy in Novi Sad. (Serbia); Department of Psychology, Faculty of Philosophy, University of Niš, Serbia (Serbia); Singapore Management University IRB (Singapore); Ethics Committee of the Center for Social and Psychological Sciences, Slovak Academy of Sciences (Slovakia); IRB at IESE Business School (Spain); Post Graduate Institute of Medicine, University of Colombo (Sri Lanka); Koç University (Turkey); New York University IRB (USA); University of Georgia IRB (USA); Institutional Review Board, University of South Carolina (USA); Cardiff School of Psychology's Research Ethics Committee (UK); Comité de Ética en Investigación/Universidad Católica del Uruguay (Uruguay); Ethics Committee of the Dept of Education and Psychology, Forman Christian College (Pakistan); Saint George's University IRB (Grenada).

### Reporting summary

Further information on research design is available in the Nature Portfolio Reporting Summary linked to this article.

## Results
### Analysis plan

Our analysis proceeds in four parts. First, we validate our measure of individualistic morality and our measures of everyday concerns across societies. Second, we present the global patterns of everyday norms. Third,

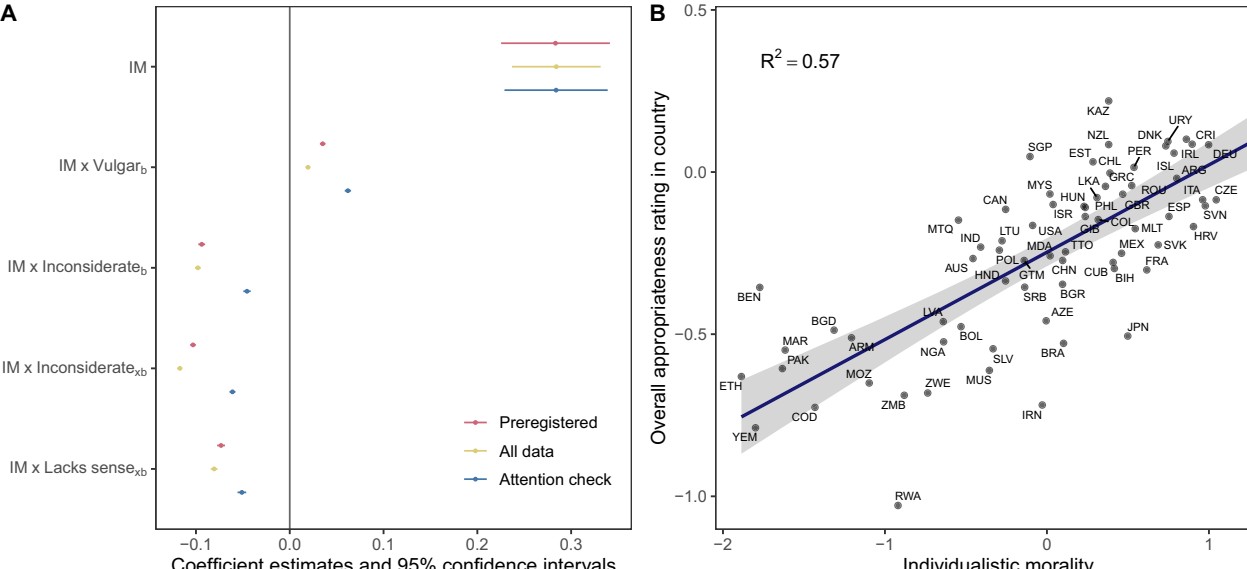

**Fig. 4 | Results on societal variation in everyday norms from a mixed-level model of appropriateness ratings. A** The graph shows the estimated coefficients for societies' scores on individualistic morality (IM) and their interactions with situation-general concerns about vulgarity (Vulgar$_b$) and inconsiderateness (Inconsiderate$_b$), and situation-specific concerns about inconsiderateness (Inconsiderate$_{xb}$) and lacking sense (Lacks sense$_{xb}$). Error bars represent 95% confidence intervals. Results are presented for data collected up to the preregistered deadline (red, $n = 547{,}170$ appropriateness ratings from 71 societies), all data

(yellow, $n = 833{,}930$ ratings from 90 societies)), and only including participants who passed the attention check (blue, $n$–515,162 ratings from 80 societies). The confidence intervals for the interaction effects become somewhat wider if random slopes are included (see Supplementary Fig. 3A). **B** Scatter plot illustrating the positive relation between a society's individualistic morality and its overall (context- and behavior-general) appropriateness rating across 71 societies. Labels are ISO codes. Gray shading indicates 95% confidence intervals.

we test our preregistered hypothesis on societal variation using a mixed-level model to analyze how individualistic morality predicts norm permissiveness. Fourth, we test our preregistered hypothesis on norm change over the last 20 years, again using a mixed-level model. Results reported in the text are based on the preregistered sample; using the full sample or only participants who passed an attention check yields similar results. Full results for all samples are presented in the figures and Supplementary Material.

## Validation of value differences

Our findings reveal significant societal variation in individualistic morality. Using the Individualistic morality scale, society-level scores ranged from −1.9 in Ethiopia to 1.3 in Sweden (see Supplementary Fig. 1 for a map). These scores demonstrated strong convergent validity, correlating highly with other value measures, such as the difference in relevance between individualizing and binding foundations ($r(69) = 0.81$, 95% CI [0.71, 0.87], $p < 0.001$), freedom-of-choice values ($r(69) = 0.90$, 95% CI [0.84, 0.94], $p < 0.001$), and strength of religious beliefs ($r(69) = −0.83$, 95% CI [−0.89, −0.74], $p < 0.001$). Crucially, an MLSEM showed excellent global fit, indicating that the single-factor structure of the Ind-Bind scale is comparable across societies (configural invariance). However, significant between-society residual variances reveal some item bias, so full measurement invariance is not established; cross-cultural comparisons should therefore be interpreted with appropriate caution.

In support of value differences between samples reflecting genuine societal differences, our measures are strongly correlated with available measures from representative samples of freedom-of-choice values ($r(59) = 0.72$, 95% CI [0.57, 0.82], $p < 0.001$) and strength of religious beliefs ($r(59) = 0.81$, 95% CI [0.70, 0.88], $p < 0.001$).

Differences between students and non-students were negligible. In the MLSEM with the measurement model, we included a regression of Individualistic morality on student status at both levels; both the within- ($b = −0.042$, 95% CI [−0.109, 0.026], $z = −1.22$, $p = 0.224$, $\beta = −0.013$) and between-society ($b = −0.218$, 95% CI [−0.835, 0.399], $z = −0.69$, $p = 0.489$, $\beta = −0.093$) effects were small and non-significant.

## Everyday concerns for 150 situated behaviors

We found extremely high cross-cultural agreement on which concerns are elicited by which behaviors. Correlations between societies with high vs. low individualistic morality were $r(148) = 0.96$, 95% CI [0.94, 0.97], $p < 0.001$ for vulgarity, $r(148) = 0.92$, 95% CI [0.90, 0.94], $p < 0.0013$ for inconsiderateness, and $r(148) = 0.88$, 95% CI [0.84, 0.91], $p < 0.001$ for lacking sense, validating our assumption that concerns are perceived similarly across cultures. For example, it is generally agreed that kissing in a job interview elicits concerns about vulgarity, that laughing out loud in the library elicits concerns about inconsiderateness, and that reading the newspaper at the movies elicits concerns about lacking sense; see Supplementary Table 8 for the full table.

We defined situational dependence as the proportion of variance in a concern not explained by the behavior itself (i.e., 1−variance explained by behavior in a two-way ANOVA). Our findings indicate very low situational dependence for vulgarity (9%), moderately high for inconsiderateness (60%), and very high for lacking sense (79%). This suggests that whether a behavior is seen as vulgar depends mostly on the behavior itself, while whether it is seen as inconsiderate or lacking sense depends heavily on the situation.

## Variation in everyday norms

Figure 2 illustrates the results of our efforts to measure everyday norms globally, presented in a color-coded matrix. In this matrix, the color of each cell represents the aggregated appropriateness rating for a specific situated behavior within a particular society. Columns represent situated behaviors, and rows represent societies. For clarity, situated behaviors are sorted by their global appropriateness rating (average value across rows), while societies are sorted by their overall appropriateness rating (average value across columns). For a larger version of this figure, which includes the names of the situated behaviors and societies, please refer to Supplementary Fig. 2.

A prominent feature of Fig. 2 is the consistent shift in colors from light to dark across each row. This pattern indicates that the relative appropriateness of various situated behaviors is similar across different societies.

Therefore, it is meaningful to speak of the average across societies as global everyday norms. Figure 3A displays the global average appropriateness ratings for each behavior across different situations. This figure illustrates the presence of both behavior-specific global norms (where some rows are generally lighter than others) and situation-specific global norms (where certain cells within each row are lighter). As expected, behavior-specific global norms are well accounted for by behavior-specific concerns regarding vulgarity and inconsiderateness ($r(13) = -0.85$, 95% CI [$-0.95$, $-0.60$], $p < 0.001$, Fig. 3B). Similarly, situation-specific global norms are well accounted for by situation-specific concerns related to inconsiderateness and lacking sense ($r(148) = -0.95$, 95% CI [$-0.96$, $-0.93$], $p < 0.001$, Fig. 3C).

### Preregistered analysis of societal variation in everyday norms

Although societal variation in everyday norms is relatively small, it may still exhibit systematic patterns. Our hypothesis regarding societal variation posits that levels of individualistic morality are expected to influence how societies differ in their overall, behavior-specific, and situation-specific appropriateness ratings. We tested this hypothesis using a mixed-level model. The key results are shown in Fig. 4A.

Figure 4A illustrates that, as hypothesized, individualistic morality is linked to higher overall appropriateness ratings ($b = 0.28$, 95% CI [$0.22$, $0.34$], $t(69.8) = 9.6$, $p < 0.001$, $\beta = 0.13$), increased appropriateness ratings for behaviors that typically elicit binding concerns ($b = 0.04$, 95% CI [$0.03$, $0.04$], $t(533525.9) = 23.0$, $p < 0.001$, $\beta = 0.02$), and decreased appropriateness ratings for behaviors that evoke individualizing concerns ($b = -0.09$, 95% CI [$-0.10$, $-0.09$], $t(533517.6) = -50.4$, $p < 0.001$, $\beta = -0.05$ for behavior specific concerns and $b = -0.10$, 95% CI [$-0.11$, $-0.10$], $t(533446.8) = -67.5$, $p = 0.000$, $\beta = -0.06$ for situation specific concens) or concerns about lacking sense in specific situations ($b = -0.07$, 95% CI [$-0.08$, $-0.07$], $t(533526.7) = -34.3$, $p < 0.001$, $\beta = -0.03$). (The model also incorporated individual-level predictors of appropriateness ratings; a full report can be found in Supplementary Table 9.) Notably, the results remained consistent whether we used only data collected up to the pre-registered deadline, included all data, or excluded participants who did not pass the attention check. Figure 4B demonstrates the strength of the main effect, showing that societies with higher individualistic morality tend to exhibit more permissive everyday norms.

### Exploratory analyses of societal variation in everyday norms

**Robustness.** We examined the robustness of our findings on societal variation in everyday norms by controlling for the economic development of societies (GDP per capita) and additional demographic variables included in the study (urban/rural background, subjective social status, and education level). The main findings regarding societal variation remained qualitatively unchanged in this analysis (see Supplementary Fig. 4). Another analysis confirmed that our findings are robust when controlling for non-independence of societies in terms of geographic and linguistic proximity (see Supplementary Fig. 5).

**Taking the common-is-moral intuition into account.** We analyzed commonness ratings of each situated behavior, aggregated at the global level. Situated behaviors that are perceived as more common are also globally rated as more appropriate, $r(148) = 0.84$, 95% CI [$0.78$, $0.88$], $p < 0.001$, across $n = 150$ situated behaviors. This association may reflect that norms impact behavior, but also that population-level behaviors impact judgments of what is appropriate through a common-is-moral intuition[26]. To examine how the latter mechanism may have impacted our results, we included global perceived commonness in our analysis of societal variation in everyday norms. The results (see Supplementary Fig. 6) indicate that ratings for uncommon behaviors were lower in societies characterized by individualistic morality ($b = 0.09$, 95% CI [$0.09$, $0.10$], $t(533497.6) = 25.7$, $p < 0.001$, $\beta = 0.02$ for behavior-specific commonness and $b = 0.13$, 95% CI [$0.12$, $0.14$], $t(533466.4) = 28.3$, $p < 0.001$, $\beta = 0.05$ for situation-specific commonness; consistent with prior

findings indicating that the common-is-moral intuition is stronger in such societies[27,28]). Furthermore, the interaction between individualistic morality and concerns about lacking sense was almost totally eliminated when commonness was included in the analysis ($b = -0.01$, 95% CI [$-0.01$, $-0.00$], $t(533432.7) = -2.8$, $p = 0.005$, $\beta = -0.00$), whereas other interactions remained relatively robust ($b = 0.04$, 95% CI [$0.04$, $0.05$], $t(533507.4) = 28.3$, $p < 0.001$, $\beta = 0.03$ for behavior-specific binding concerns; $b = -0.09$, 95% CI [$-0.09$, $-0.08$], $t(533511.1) = -46.5$, $p < 0.001$, $\beta = -0.04$ for behavior-specific individualizing concerns; and $b = -0.06$, 95% CI [$-0.06$, $-0.05$], $t(533426.0) = -24.9$, $p < 0.001$, $\beta = -0.03$ for situation-specific individualizing concerns). These findings suggest that it is via the common-is-moral intuition that concerns about lacking sense impact societal differences in norms.

**The actual vs. perceived relevance of everyday concerns.** We estimated the relevance of everyday concerns in every society by regressing appropriateness ratings of 150 situated behaviors on the three measures of the concerns each situated behavior elicits. After reversing the sign, the coefficient for, say, inconsiderateness is a measure of how much appropriateness ratings drop for a unit increase in inconsiderateness concerns. In every society, we also aggregate participants' ratings of how relevant each concern is for them when deciding whether a behavior is appropriate. These two kinds of relevance measures are correlated across 71 societies: $r(69) = 0.37$ (95% CI [$0.15$, $0.55$], $p = 0.002$) for concerns about vulgarity, $r(69) = 0.84$ (95% CI [$0.75$, $0.85$], $p < 0.001$) for concerns about inconsiderateness, and $r(69) = 0.25$ (95% CI [$0.01$, $0.45$], $p = 0.039$) for concerns about lacking sense. Thus, the impact of various everyday concerns on norms is broadly consistent with people's perceptions of how relevant each concern is to their normative judgments.

### Global change in everyday norms over 20 years

Data on appropriateness ratings for twelve behaviors across 10 situations in 26 societies were collected at two time points: first from 2000 to 2003 by Gelfand and colleagues, and again from 2023 to 2024 in the current study. The pattern of norm change for the 120 situated behaviors was highly consistent across the 22 societies, with a Cronbach's $\alpha$ of 0.94. Figure 5 illustrates the changes in global average appropriateness ratings from the first wave to the second.

A striking feature of Fig. 5 is the remarkable similarity in global norms between the two studies. Behaviors deemed inappropriate twenty years ago continue to be viewed as such today, and the same applies to those considered appropriate. This indicates that there has been little change in everyday norms over the past twenty years. However, despite their small magnitude, the changes that have occurred have been highly consistent across societies. For instance, in nearly every society, norms have become more restrictive regarding arguing and flirting, while norms for crying, eating, listening to music on headphones, and reading the newspaper have become more permissive. See Table 1 for further details.

### Preregistered analysis of global change in everyday norms

Systematic changes in norms may arise from shifts in underlying values. Our hypothesis posited that global everyday norms would evolve over time in a manner similar to their variation with the individualistic morality of different societies. This hypothesis was based on the assumption that global values are changing toward more individualistic morality, with liberty and care becoming increasingly relevant and purity less relevant. We tested this hypothesis using a mixed-level model of appropriateness ratings, coding time as 0 for the 2000–2003 data and 1 for the current data. Figure 6 presents a summary of the key results from this analysis. For a comprehensive report of the results, please refer to Supplementary Table 10.

Figure 6A indicates that everyday norms have generally become more permissive ($b = 0.03$, 95% CI [$0.01$, $0.05$], $t(11490.7) = 2.9$, $p = 0.004$, $\beta = 0.01$), but more restrictive specifically for behaviors that raise concerns about inconsiderateness ($b = -0.15$, 95% CI [$-0.16$, $-0.15$], $t(741434.6) = -52.2$, $p < 0.001$, $\beta = -0.04$). These findings are in line with

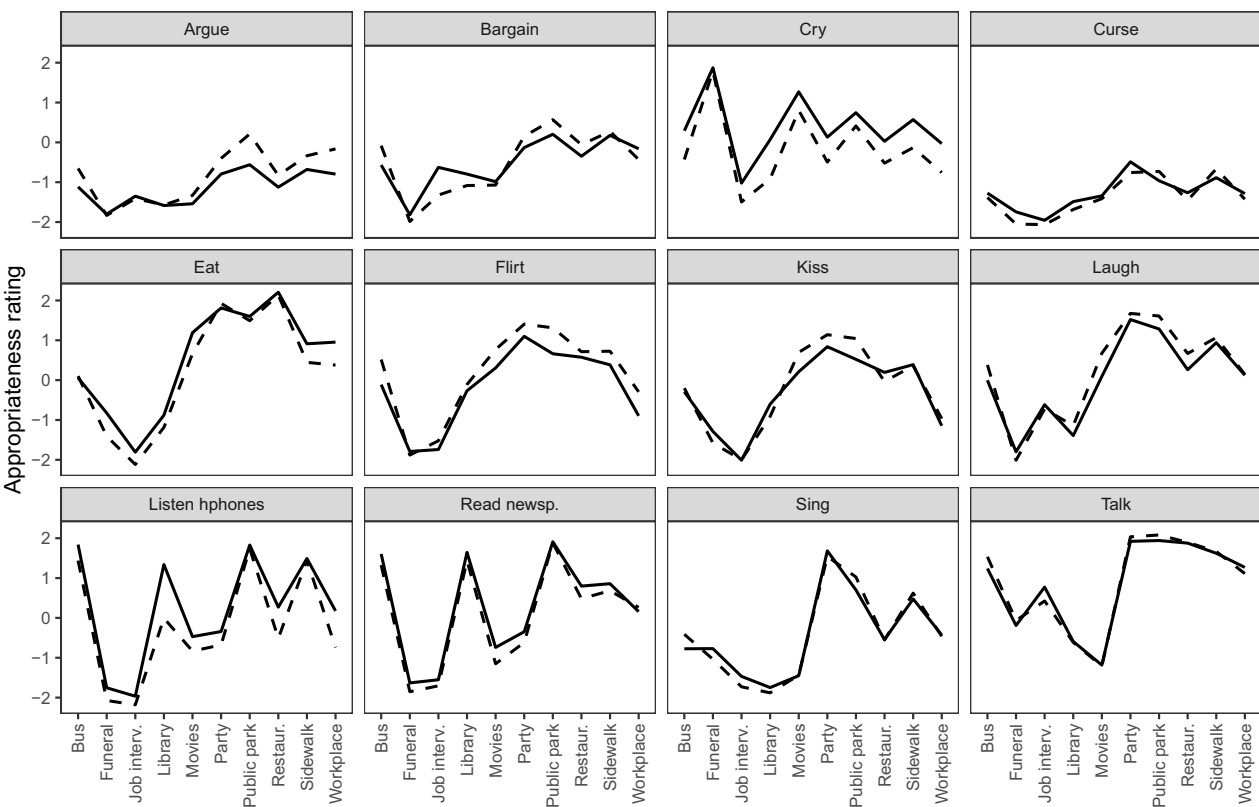

**Fig. 5 | The global average change in everyday norms.** Every panel shows appropriateness ratings of a specific behavior in ten different situations at two times: 2000–2003 (dashed lines) and 2023–2024 (solid lines). Ratings are aggregated across 22 societies (but only 21 for laughing, because Israel missed 2000–2003 data for laughing at a party).

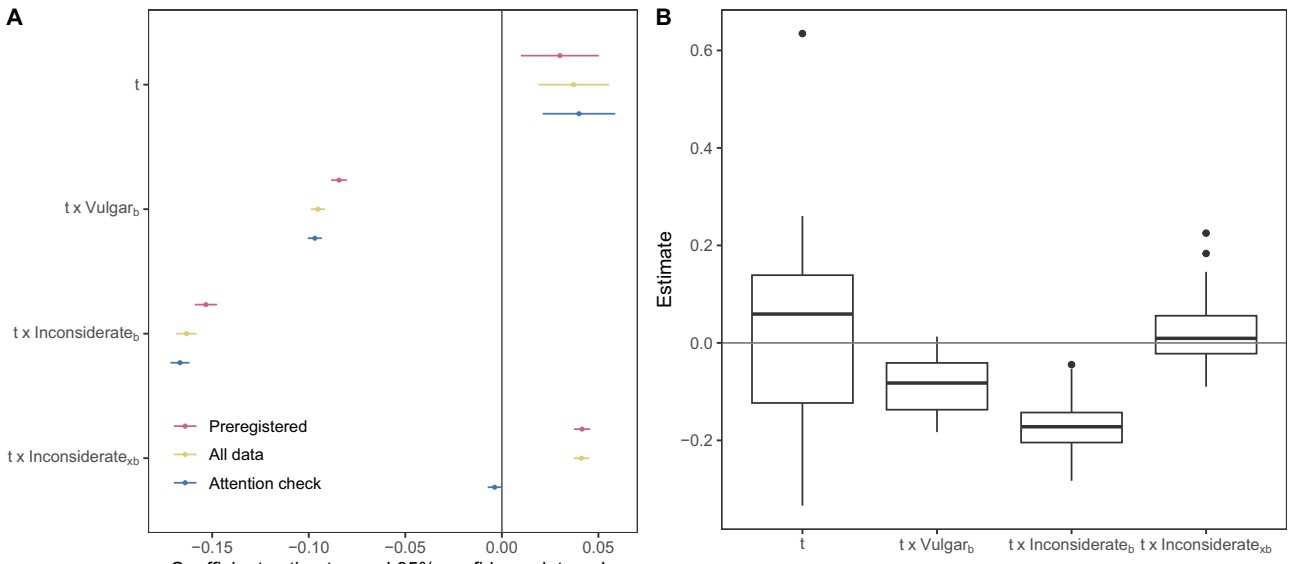

**Fig. 6 | Results on global change in everyday norms from a mixed-level model of appropriateness ratings. A** The graph shows the estimated coefficients for time and its interactions with situation-general concerns about vulgarity and inconsiderateness, and situation-specific concerns about inconsiderateness. Error bars represent 95% confidence intervals. Results are presented for data collected up to the preregistered deadline (red, $n$ = 869,180 appropriateness ratings from 22 societies), all data (yellow, $n$ = 929,143 appropriateness ratings from 26 societies), and only

including participants who passed the attention check (blue, $n$ = 856,692 appropriateness ratings from 26 societies). The confidence intervals for the interaction effects become somewhat wider if random slopes are included (see Supplementary Fig. 3B). **B** Boxplots of the coefficients for time and its interactions with concerns estimated separately in each of the 22 societies. Dots represent societies, boxes indicate the first and third quartiles, and the bold lines indicate medians.

our hypotheses. Norms have also become more restrictive specifically for behaviors that provoke concerns about vulgarity ($b$ = -0.08, 95% CI [−0.09, −0.08], $t(741335.1) = −40.7$, $p < 0.001$, $\beta$ = -0.03), but we hypothesized the opposite. We discuss a possible explanation for this unexpected finding

below. Norms for behaviors that elicit concerns about inconsiderateness in a specific situation were estimated to have become slightly more permissive ($b$ = 0.04, 95% CI [0.04, 0.05], $t(741311.3) = 19.3$, $p < 0.001$, $\beta$ = 0.01); however, this finding is not robust, as it was reversed when the attention

check was applied. Most of these norm changes are consistent across societies, as illustrated in Fig. 6B.

## Discussion

This study mapped everyday norms across 90 societies, revealing how they vary across behaviors, situations, societies, and time. Our findings show that despite this complexity, much of the variation can be parsimoniously explained by the interplay between a single cultural value dimension—individualistic versus binding morality—and a few core everyday concerns.

A key finding is that societies vary considerably in their level of individualistic morality, yet show remarkable agreement on which concerns (vulgarity, inconsiderateness, lacking sense) are elicited by specific situated behaviors. This supports the psychological validity of these distinctions and provides the foundation for our model: norms differ between societies not because the meaning of behavior changes, but because the moral weight given to the concerns that behaviors elicit varies with societal values. Societies with higher individualistic morality have more permissive norms overall (reflecting liberty) but are stricter about behaviors that are inconsiderate (reflecting care) and odder (reflecting the common-is-moral intuition), while being more tolerant of behaviors considered merely vulgar (reflecting less emphasis on purity).

### Interpretation in broader theoretical context

While we used the individualizing/binding dimension from MFT[17,24] as our primary lens, our findings can be interpreted within the broader landscape of moral psychology. For instance, our results complement the cultural dimension of tightness–looseness[8,9]. Societies with higher individualistic morality and more permissive norms can be seen as looser cultures. Our model helps specify the moral content of this variation, showing that looseness is not uniform but is instead structured by the type of moral concern a behavior evokes.

Furthermore, our finding that norms for vulgar behaviors (e.g., flirting, cursing) became *less* permissive over time was unexpected from a simple MFT perspective, which would predict that declining emphasis on purity should lead to more permissive norms. However, this result can be plausibly explained with the Theory of Dyadic Morality (TDM). While we conceptualized vulgarity as a non-consequentialist Purity violation, it can also be construed as a form of indirect harm that causes offense to others[12]. If societies are indeed becoming more considerate, this increased sensitivity to harming or offending others could lead to stricter judgments against public displays of vulgarity, even as personal disapproval of such acts declines. This suggests a fruitful area for future research on the interaction between different moral concerns.

Our study also sheds light on the nature of situationism across cultures[53–55]. We found that the impact of situationally-dependent concerns (inconsiderateness and lacking sense) on norms was stronger in more individualistic societies. This suggests that situationism—the tendency to consider the situation when judging behavior—may itself vary systematically across societies depending on which underlying moral concerns are prioritized. This theory warrants further exploration in future studies.

### Limitations

A notable feature of this survey experiment is the independent manipulation of behaviors and situations, allowing us to examine the situational dependence of everyday concerns and norms. Another major strength of the study is its large and culturally diverse sample. The use of convenience samples is, of course, a limitation for society-level estimates. However, these samples included both students and non-students and were shown to effectively represent the values of their respective societies. Demographic shifts could be a confound for the change analysis, but our model controlled for the basic demographic variables age and gender.

Another limitation is that the change analysis only relies on two data points 20 years apart, hence it cannot reveal whether the trajectory of change was gradual, recent, or fluctuating. More frequent time-series data is needed to map these dynamics.

The framework outlining three primary concerns about everyday behavior is not exhaustive. It may be extended to include other concerns, such as authority or loyalty, that are undoubtedly relevant in hierarchical or intergroup contexts. Such contexts were not studied here but represent an avenue for future work.

Our study operationalized norms solely through aggregated appropriateness ratings. It is known that norm enforcement—such as confrontation, ostracism, and gossip—varies across societies[56]. How norm enforcement varies across everyday behaviors and situations remains an open question for future research.

Finally, our study introduced a tool, the Ind-Bind scale, to measure the relative priority societies' place on individualizing versus binding concerns. It is distinct from the Moral Foundations Questionnaire in that it explicitly asks people to compare violations of different types of concerns instead of evaluating each concern separately. While both formats have advantages and disadvantages, we found that both scales yielded very similar results.

## Conclusion

In conclusion, this study uncovers a global grammar of everyday norms. We find striking similarities in what people around the world frown upon, with norms varying far more across situations than across societies. This variation is systematically structured by a few everyday concerns whose relevance is weighted by a single dimension of cultural values. An implication of our findings is that if someone moves to a different society and questions the normativity of a mundane behavior in a specific situation, the answer is likely to be similar to what they experienced in their original society—however, if relocating to a less individualistic society, they may find the norms to be somewhat stricter, particularly for behaviors perceived as vulgar, though less so for those deemed inconsiderate. Moreover, everyday norms are not static; they appear to be changing in consistent ways globally, suggesting a shared trajectory of cultural evolution. Our findings provide insights into the interplay of human universals and cultural differences that shape social life across the globe.

## Data availability

All data and materials generated and/or analyzed in this study, including the raw data underlying the figures and tables, are available at OSF (https://osf.io/sh4rb/, https://doi.org/10.17605/OSF.IO/SH4RB).

## Code availability

The R syntax for all analyses are available at OSF (https://osf.io/sh4rb/, https://doi.org/10.17605/OSF.IO/SH4RB).

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

## Acknowledgements

Michele Gelfand provided valuable comments on the study design and the manuscript. Funders that supported this research include the Knut and Alice Wallenberg Foundation (grant no. 2022.0191; Pontus Strimling), the Higher School of Economics Basic Research Program (Ekaterina Nastina and Natalia Soboleva), NextGenerationEU (contract no. BG-RRP-2.004-0007-C01; Kristina Stoyanova), the John Templeton Foundation (grant no. 62631; Robert M. Ross), the Australian Research Council (grant no. DP180102384; Robert M. Ross), the Youth Innovation Promotion Association, Chinese Academy of Sciences (grant no. 2023095; Junhui Wu), Open University of Israel (grant no. 48766; Ravit Nussinson), JSPS KAKENHI (grant no. JP21K02983; Toko Kiyonari); Narodowe Centrum Nauki (grant no. 2019/35/B/HS6/01421; Katarzyna Growiec), National Research and Development Fund (grant no. NKFIH-OTKA-K 135963; Marta Fulop), Australian National University (Samantha K. Stanley), the Swedish Research Council (grant no. 2023-01306; Giulia Andrighetto), Shota Rustaveli National Science Foundation (grant no. FR-22-15319; Vladimer Gamsakhurdia), The Science Fund of the Republic of Serbia (grant no. 7744418; Bojana M. Dinić), CIS—Centro de Investigação e Intervenção Social (CIS-ISCTE) through funds allocated by the Portuguese Foundation for Science and Technology (FCT) (grant no. UIDB/03125/2020, https://doi.org/10.54499/UIDB/03125/2020: Ricardo B. Rodrigues), Czech Science Foundation (grant no. GA23-06170S; Sylvie Graf and Martina Hřebíčková), the Ministry of Science, Technological Development and Innovations of the Republic of Serbia (contract no. 451-03-66/2024-03; Ivana Pedović), Linköping University (Maria Luisa Mendes Teixeira). The funders had no role in study design, data collection and analysis, decision to publish, or preparation of the manuscript.

## Author contributions

K.E. designed the study and wrote the paper. P.S. co-designed the study and assisted with the writing. I.V. co-designed the study and performed the statistical analysis and visualizations. B.S. assisted with the writing and collected data. Mi.P. coordinated the international data collection and collected data. Khal.A., N.A., Alis.A., H.A., M.A., Khat.A., Y.A., Alb.A., P.A., Gi.A., Giz.A., J.A., C.A., Jon.B., D.B., Jus.B., Bi.B., A.B., E.Ber., S.N.B., M.Bj, S.B., P.B., E.Boš., Y.B., M.Br, K.B., H.T.T.B., T.C., Y.C., M.K.C., H.C., Car.C., D.C., Chr.C., A.C., P.d.Z., Z.D., B.M.D., S.D., R.E., J.B.E., I.E., H.E., X.F., C.F., E.F., M.F., V.G., M.A.G.J., R.G., Ali.G., C.M.H.D.G., B.G., And.G., Sy.G., Ani.G., Ka.G., B.W.H., Ge.H., S.P.H., N.H., M.H.H.M.A., E.H., P.H., Gi.H., An.H., Ma.H., B.C.H., S.H., M.H., J.A.H., M.I., D.I., H.L.H.J., Z.K., Ha.K., I.K., Ga.K., Ke.K., N.K., J.B.K., J.K., T.K., M.Ko., S.Ko., B.K., L.L., M.L., B.L., Z.L., L.M.L., Y.L., K.L., M.J.L., M.E.L.R., W.L., E.M., Mo.M., B.A.M., N.M.M., I.M., M.L.M.T., J.P.M.T., L.L.M., S.N.M., B.Mo., N.Mu., H.N., E.N., P.N., D.N., O.N., M.N., P.Nt., R.N., R.Nu., M.O., N.G.O., I.E.O., P.P., D.P.A., Md.P., G.L.P., I.P., P.P.d.L., L.R.P.F., N.P., J.R.P., A.P., A.Q., J.L.R., R.B.R., J.D.R., S.Ro., R.M.R., N.Ro., Sel.S., A.S.M., Sne.S., Sar.S., N.S., Dan.S., Sam.S., K.S., Dro.S., Ko.T., J.T., Hab.T., Han.T., T.U., E.M.U., R.W., Y.W., J.W., B.M.Y., E.Y. and K.H.Y. collected data. P.A.M.V.L. assisted with the study design and writing. All authors reviewed the manuscript.

## Funding

## Competing interests

The authors declare no competing interests.

## Additional information

Kimmo Eriksson [1,2] ✉, Pontus Strimling [1,3,4], Irina Vartanova[1,3], Brent Simpson [5], Minna Persson [1], Khalid Ahmed Abdi[6], Neta Ad[7], Alisher Aldashev[8], Habib Mohammad Ali [9], Maurizio Alì [10], Khatai Aliyev [11,12], Yasser M. H. A. Alrefaee [13], Alberth Estuardo Alvarado Ortiz [14], Per A. Andersson [15], Giulia Andrighetto[1,4,16], Gizem Arikan [17], John Jamir Benzon R. Aruta[18], Christian Lutete Ayikwa [19], Jonatan Baños-Chaparro[20], Davide Barrera [21,22], Justina Barsyte[23,24], Birzhan Batkeyev[8], Azma Batool[25], Elizaveta Berezina [26], Stéphanie Ngandu Bimina [27], Marie Björnstjerna[28], Sheyla Blumen [29], Paweł Boski[30], Eva Boštjančič [31], Yap Boum II[32], Marie Briguglio [33], Kagonbe Bruno[34], Huyen Thi Thu Bui[35], Tomás Caycho-Rodríguez [36], Yanyan Chen[37,38], Manase Kudzai Chiweshe [39], Hoon-Seok Choi[40], Carlos C. Contreras-Ibáñez [41], Dinka Čorkalo [42], Christian E. Cruz-Torres [43], Andrea Czakó [44,45], Piyanjali de Zoysa [46], Zsolt Demetrovics [44,45,47], Bojana M. Dinić [48], Saša Drače [49], Rita W. El-Haddad[50]

Jan B. Engelmann [51], Ignacio Escudero Pérez[52], Hyun Euh [53], Xia Fang [54], Celine Frank [55,56], Esteban Freidin[57], Marta Fulop [58,59], Vladimir Gamsakhurdia[60], Mauro Alberto García Jiménez [61], Ragna B. Gardarsdottir [62], Alin Gavreliuc [63], Colin Mathew Hugues D. Gill[64], Biljana Gjoneska [65], Andreas Glöckner [55,66], Sylvie Graf [67], Ani Grigoryan[68], Katarzyna Growiec [30], Brian W. Haas [69], Geoffrey Haddock[70], Stavros P. Hadjisolomou [50], Nina Hadžiahmetović [49], Mohammad Hosein Haji Mohammad Ali [71], Eemeli Hakoköngäs [72], Peter Halama [73], Given Hapunda[74], Andree Hartanto [75], Mahsa Hazrati [71], Boris Christian Herbas-Torrico [76], Szilárd Holka [57,77], Martina Hřebíčková [67], John A. Hunter[78], Moudachirou Ibikounle [79], Dzintra Ilisko[80], Harpa Lind Hjördísar Jónsdóttir [62], Zivile Kaminskiene[23,24], Hansika Kapoor [81], Iva Kapović[42], Gassemi Karim [82], Kerry Kawakami[83], Narine Khachatryan[68], Julian B. Kirschner[51], Jonah Kiruja[6], Toko Kiyonari [84], Michal Kohút [85], Shazia Kousar[86], Besnik Krasniqi[87], Ludovic Lado[34], Miguel Landa-Blanco [88], Barbara Landon[89], Žan Lep [31,90], Lisa M. Leslie[91], Yang Li[92], Kadi Liik [93], Ming-Jen Lin[94], Marlon Elías Lobos Rivera[95], Wilson López-López [96], Edona Maloku [97], Mohona Mandal[5], Bernardo Ananias Manhique[98], Nathan Mpeti Mbende[19], Imed Medhioub [99], Maria Luisa Mendes Teixeira [100], J. Paola Merchán Tamayo[5], Linda Lila Mohammed[101], Schontal N. Moore [102], Bahar Moraligil[103], Nijat Muradzada [11,104], Herwin Nanda[32], Ekaterina Nastina[105], Pegah Nejat [71], Daniel Nettle [106,107], Orlando Julio Andre Nipassa [98], Martin Noe-Grijalva[108], Pie Ntampaka [109], Rodrigue Ntone[32], Ravit Nussin0n[7], Milan Oljača[48], Nneoma G. Onyedire [110], Ike E. Onyishi[110], Penny Panagiotopoulou[111], Daybel Pañellas Alvarez[112], Md. Shahin Parvez [113], Gian Luca Pasin[16], Ivana Pedović[114], Pablo Pérez de León[52], Lorena R. Perez Floriano [115], Nada Pop-Jordanova[65], Jose Roberto Portillo[14], Angela Potang [116], Adolfo Quesada-Román [117], Jana L. Raver [118], Ricardo B. Rodrigues[119], Juan Diego Rodríguez-Romero [96], Sara Romanò [21], Robert M. Ross [120], Nachita Rosun[121], Selka Sadiković[48], Alvaro San Martin [122], Snežana Smederevac [48], Sarah Jane Smith[70], Natalia Soboleva[105], Daniel Erena Sonessa[123], Samantha K. Stanley[124], Kristina Stoyanova[125], Drozdstoy Stoyanov [125], Kosuke Takemura [126], John Thøgersen [127], Habib Tiliouine[128], Hans Tung[129], Tungalag Ulambayar [130], Elze Marija Uzdavinyte[23,24], Randall Waechter[131], Yi-Ting Wang[132], Junhui Wu [37,38], Brice Martial Yambio [133,134], Eric Yankson[135], Kuang-Hui Yeh [136] & Paul A. M. Van Lange [137]

[1]Institute for Futures Studies, Stockholm, Sweden. [2]School of Education, Culture and Communication, Mälardalen University, Västerås, Sweden. [3]Department of Women's and Children's Health, Uppsala University, Uppsala, Sweden. [4]Institute for Analytical Sociology, Linköping University, Norrköping, Sweden. [5]Department of Sociology, University of South Carolina, Columbia, SC, USA. [6]Center for Research, Publication and Community Service, University of Hargeisa, Hargeisa, Somaliland. [7]Department of Education and Psychology, The Open University of Israel, Raanana, Israel. [8]International School of Economics, Kazakh-British Technical University, Almaty, Kazakhstan. [9]Department of Media Studies and Journalism, University of Liberal Arts, Dhaka, Bangladesh. [10]Institut national supérieur du professorat et de l'éducation de Martinique, Université des Antilles, Fort-de-France, Martinique, France. [11]UNEC Empirical Research Center, Azerbaijan State University of Economics (UNEC), Baku, Azerbaijan. [12]Department of Economics, Indiana University Bloomington, Bloomington, IN, USA. [13]Albaydha University, Al Bayda, Yemen. [14]Department of Applied Mathematics, Universidad Galileo, Guatemala City, Guatemala. [15]JEDILab, Department of Behavioral Sciences and Learning, Linköping University, Linköping, Sweden. [16]Institute for Cognitive Sciences and Technologies, Rome, Italy. [17]Trinity College Dublin, Dublin, Ireland. [18]Department of Psychology, De La Salle University, Manila, Philippines. [19]Département d'Administration des Affaires, Faculté d'Administration des Affaires et Sciences Economiques, Université Protestante au Congo, Kinshasa, Democratic Republic of the Congo. [20]Universidad Privada Norbert Wiener, Vicerrectorado de Investigación, Lima, Peru. [21]Department of Culture, Politics and Society, University of Turin, Turin, Italy. [22]Collegio Carlo Alberto, Turin, Italy. [23]AdCogito Institute for Advanced Behavioral Research, Vilnius, Lithuania. [24]Center for Economic Expertise, Vilnius University, Vilnius, Lithuania. [25]Forman Christian College (A Chartered University), Lahore, Pakistan. [26]Department of Psychology, Sunway University, Selangor Darul Ehsan, Malaysia. [27]Département de Gestion d'Entreprises, Institut Supérieur Pédagogique de la Gombe, Kinshasa, Democratic Republic of the Congo. [28]Department of Psychology, Lund University, Lund, Sweden. [29]Department of Psychology, Pontifical Catholic University of Peru, Lima, Peru. [30]Faculty of Psychology, SWPS University, Warsaw, Poland. [31]Social Psychology and Policy Lab, Department of Psychology, Faculty of Arts, University of Ljubljana, Ljubljana, Slovenia. [32]Homegrown Solutions For Health, Research Department, Yaoundé, Cameroon. [33]Department of Economics, Faculty of Economics, Management and Accountancy, University of Malta, Msida, Malta. [34]CEFOD Business School, N'Djamena, Chad. [35]Faculty of Psychology and Education, Hanoi National University of Education, Hanoi, Vietnam. [36]Facultad de Psicología, Universidad Científica del Sur, Lima, Peru. [37]CAS Key Laboratory of Behavioral Science, Institute of Psychology, Chinese Academy of Sciences, Beijing, China. [38]Department of Psychology, University of Chinese Academy of Sciences, Beijing, China. [39]University of Zimbabwe, Harare, Zimbabwe. [40]Sungkyunkwan University, Seoul, South Korea. [41]Departamento de Sociología, Universidad Autónoma Metropolitana, Mexico City, Mexico. [42]Department of Psychology, University of Zagreb, Zagreb, Croatia. [43]Department of Psychology, University of Guanajuato, Guanajuato, Mexico. [44]Centre of Excellence in Responsible Gaming, University of Gibraltar, Gibraltar, Gibraltar. [45]Institute of Psychology, ELTE Eötvös Loránd University, Budapest, Hungary. [46]Faculty of Medicine, University of Colombo, Colombo, Sri Lanka. [47]Flinders University Institute for Mental Health and Wellbeing, College of Education, Psychology and Social Work, Flinders University, Bedford Park, SA, Australia. [48]Department of Psychology, Faculty of Philosophy, University of Novi Sad, Novi Sad, Serbia. [49]Department of Psychology, Faculty of Philosophy, University of Sarajevo, Sarajevo, Bosnia and Herzegovina. [50]Department of Social and Behavioral Sciences, American University of Kuwait, Salmiya, Kuwait. [51]Amsterdam School of Economics, University of Amsterdam, Amsterdam, The Netherlands. [52]Universidad Católica del Uruguay, Montevideo, Uruguay. [53]Gies College of Business, University of Illinois Urbana-Champaign, Champaign, IL, USA. [54]Department of Psychology and Behavioral Sciences, Zhejiang University, Zhejiang, China. [55]Department of Psychology, University of Cologne, Cologne, Germany. [56]Faculty of Psychology, TU Dresden, Dresden, Germany. [57]Instituto de Investigaciones Económicas y Sociales del Sur (IIESS), UNS-CONICET, Bahía Blanca, Argentina. [58]HUN-REN Institute of Cognitive Neuroscience and Psychology, Research Centre of Natural Sciences, Budapest, Hungary. [59]Karoli Gaspar University of the Reformed Church in Hungary, Institute of Psychology, Budapest, Hungary. [60]Ivane Javakhishvili Tbilisi State University, Tbilisi, Georgia. [61]Facultad de Psicología, Universidad Nacional Autónoma de México, Mexico City, Mexico. [62]Faculty of Psychology, University of Iceland, Reykjavík, Iceland. [63]Department of Psychology, West University of Timisoara, Timişoara, Romania. [64]Universal College Bangladesh, Dhaka, Bangladesh. [65]Macedonian Academy of Sciences and Arts, Skopje, North Macedonia. [66]Max Planck Institute for Research on Collective Goods, Bonn, Germany. [67]Institute of Psychology, Czech Academy of Sciences, Brno, Czechia. [68]Department of Personality Psychology, Yerevan State University, Yerevan, Armenia. [69]Department of Psychology, University of Georgia, Athens, GA, USA. [70]School of Psychology, Cardiff

University, Cardiff, UK. [71]Department of Psychology, Faculty of Education and Psychology, Shahid Beheshti University, Tehran, Iran. [72]Discipline of Social Psychology, University of Helsinki, Helsinki, Finland. [73]Center for Social and Psychological Sciences, Slovak Academy of Sciences, Bratislava, Slovakia. [74]University of Zambia, School of Humanities and Social Sciences, Department of Psychology, Lusaka, Zambia. [75]School of Social Sciences, Singapore Management University, Singapore, Singapore. [76]Tecnologico de Monterrey, Department of Industrial Engineering, Guadalajara, Mexico. [77]Department of Psychiatry and Psychotherapy, Semmelweis University, Budapest, Hungary. [78]Department of Psychology, University of Otago, Dunedin, New Zealand. [79]Centre de Recherche pour la lutte contre les Maladies Infectieuses Tropicales (CReMIT/TIDRC), Université d'Abomey-Calavi, Abomey-Calavi, Benin. [80]Institute of Humanities and Social Sciences, Daugavpils University, Daugavpils, Latvia. [81]Department of Psychology, Monk Prayogshala, Mumbai, India. [82]ENCG - University Hassan II, Casablanca, Morocco. [83]Department of Psychology, York University, Toronto, Canada. [84]School of Social Informatics, Aoyama Gakuin University, Kanagawa, Japan. [85]Faculty of Philosophy and Arts, Trnava University, Trnava, Slovakia. [86]Department of Economics, Lahore College for Women University Lahore, Lahore, Pakistan. [87]Department of Management, University of Prishtina, Prishtina, Kosovo. [88]School of Psychological Sciences, National Autonomous University of Honduras, Tegucigalpa, Honduras. [89]Windward Islands Research and Education Foundation, St. Georges University, True Blue, West Indies, Grenada. [90]Centre for Applied Epistemology, Educational Research Institute, Ljubljana, Slovenia. [91]Stern School of Business, New York University, New York, NY, USA. [92]School of Informatics, Nagoya University, Nagoya, Japan. [93]Tallinn University, Tallinn, Estonia. [94]Department of Economics, National Taiwan University, Taipei, Taiwan. [95]Universidad Tecnológica de El Salvador, San Salvador, El Salvador. [96]Pontificia Universidad Javeriana, Bogotá, Colombia. [97]Independent Researcher, Prishtina, Kosovo. [98]Eduardo Mondlane University, Maputo, Mozambique. [99]Department of Finance, Imam Muhammad Ibn Saud Islamic University (IMSIU), Riyadh, Saudi Arabia. [100]Universidade Presbiteriana Mackenzie, São Paulo, Brazil. [101]Institute of Criminology and Public Safety, University of Trinidad and Tobago, Port of Spain, Trinidad and Tobago. [102]School of Education, The University of the West Indies, Mona, Jamaica. [103]Loughborough Business School, Loughborough University, Loughborough, UK. [104]School of Social and Political Sciences, University of Glasgow, Glasgow, UK. [105]Ronald F. Inglehart Laboratory for Comparative Social Research, Higher School of Economics, Moscow, Russia. [106]Institut Jean Nicod, Département d'études cognitives, ENS-PSL, CNRS, Paris, France. [107]Department of Social Work, Education and Community Wellbeing, Northumbria University, Newcastle, UK. [108]Escuela de Psicología, Universidad César Vallejo, Trujillo, Peru. [109]Department of Veterinary Medicine, University of Rwanda, Nyagatare city, Rwanda. [110]Department of Psychology, University of Nigeria, Nsukka, Nigeria. [111]Department of Educational Sciences and Social Work, University of Patras, Patras, Greece. [112]Facultad de Psicologìa, Universidad de La Habana, Habana, Cuba. [113]Department of Sociology, First Capital University of Bangladesh, Chuadanga, Bangladesh. [114]Department of Psychology, Faculty of Philosophy, University of Niš, Niš, Serbia. [115]Facultad de Administración y Economía, Universidad Diego Portales, Santiago, Chile. [116]Department of Psychology, Moldova State University, Chisinau, Moldova. [117]School of Geography, Universidad de Costa Rica, San José, Costa Rica. [118]Smith School of Business, Queen's University, Kingston, Canada. [119]Instituto Universitário de Lisboa ISCTE-IUL, CIS, Lisbon, Portugal. [120]Department of Philosophy, Macquarie University, Sydney, Australia. [121]Brunel University, London, UK. [122]IESE Business School, University of Navarra, Madrid, Spain. [123]School of Journalism and Communication, Addis Ababa University, Addis Ababa, Ethiopia. [124]UNSW Institute for Climate Risk & Response, University of New South Wales, Sydney, Australia. [125]Research Institute at Medical University of Plovdiv, Plovdiv, Bulgaria. [126]Faculty of Economics, Shiga University, Hikone, Japan. [127]Department of Management, Aarhus University, Aarhus, Denmark. [128]Faculty of Social Sciences, University of Oran2 Mohamed Ben Ahmed, Oran, Algeria. [129]Department of Political Science, National Taiwan University, Taipei, Taiwan. [130]Zoological Society of London, London, UK. [131]Caribbean Center for Child Neurodevelopment at WINDREF, True Blue, West Indies, Grenada. [132]Department of Political Science, National Cheng Kung University, Tainan, Taiwan. [133]Institut Pasteur de Bangui, Bangui, Central African Republic. [134]Faculté des Sciences de l'Université de Bangui, Bangui, Central African Republic. [135]Namibia University of Science and Technology, Windhoek, Namibia. [136]Institute of Ethnology, Academia Sinica, Taipei, Taiwan. [137]VU University, Amsterdam, The Netherlands.

✉ e-mail: kimmo.eriksson@mdu.se

