## [Transparent Peer Review file · Communications Psychology]

Everyday norms have become more permissive over time and vary across cultures

Corresponding Author: Professor Kimmo Eriksson

Version 0:

Decision Letter:

Dear Professor Eriksson,

Your manuscript titled "How everyday norms vary across behaviors, situations, societies, and time" has now been seen by our reviewers, whose comments appear below. In light of their advice I am delighted to say that we are happy, in principle, to publish a suitably revised version in Communications Psychology.

We therefore invite you to revise your paper one last time to address the remaining concerns of our reviewers and a list of editorial requests. At the same time we ask that you edit your manuscript to comply with our format requirements and to maximise the accessibility and therefore the impact of your work.

EDITORIAL REQUESTS:

SUBMISSION INFORMATION:

OPEN ACCESS:

* **CODE AVAILABILITY:** All Communications Psychology manuscripts must include a section titled "Code Availability" at the

end of the methods section. We require that the custom analysis code supporting your conclusions is made available in a publicly accessible repository at this stage; please choose a repository that generates a digital object identifier (DOI) for the code; the link to the repository and the DOI must be included in the Code Availability statement. Publication as Supplementary Information will not suffice.

*** DATA AVAILABILITY:**

Link Redacted

Best regards,

Jennifer Bellingtier

Jennifer Bellingtier, PhD
Senior Editor
Communications Psychology

REVIEWERS' EXPERTISE:

Reviewer #2 social and cultural psychology
Reviewer #3 moral psychology

REVIEWERS' COMMENTS:

Reviewer #2 (Remarks to the Author):

I think that the authors have done a fine job addressing my comments in the revision.

Reviewer #3 (Remarks to the Author):

I really commend the authors for their thoughtful and thorough revisions. They responded carefully to each of my concerns, and I feel the paper is now much better situated within the broader context of research on moral psychology beyond MFT alone. It's clear they put great care into strengthening the manuscript, and I believe it will make a wonderful contribution to the field.

Warmly,

Kyle Fiore Law

Response to Reviewers

To: The Editors, *Communications Psychology*

Subject: Response to reviews for manuscript "How everyday norms vary across behaviors, situations, societies, and time."

Dear Dr. Schiffer and the editorial team,

Thank you for the opportunity to revise our manuscript, "How everyday norms vary across behaviors, situations, societies, and time" for submission to *Communications Psychology*. We are grateful to the four reviewers for their feedback. We have undertaken a substantial revision to address all the concerns raised and streamline the text for easier reading. Below, we provide a detailed, point-by-point response to each of the reviewers' comments. We include direct quotes from the revised text to illustrate the changes we have made.

Thank you again for your consideration.

Sincerely, Professor K. Eriksson and co-authors

Reviewer #1

We thank Reviewer #1 for their detailed, insightful, and very helpful comments. We have addressed each point below.

Introduction

- Comment 1: Missing theoretical context (MFT not the only theory).
 - Response: We fully agree with this crucial point. To address it, we have substantially revised the Introduction to situate our work within the broader theoretical landscape of moral psychology. We now explicitly introduce several other influential frameworks before justifying our use of MFT. Specifically, we have added the following paragraph: "*In this paper, we propose that variation in everyday norms can be understood through the lens of societal moral values. The field of moral psychology offers several influential frameworks for understanding moral judgment and its cultural variation. For instance, the Theory of Dyadic Morality posits that all moral judgments are rooted in a universal template of perceived harm⁹, while Morality-as-Cooperation theory suggests that morality evolved as a suite of distinct solutions to promote cooperation¹⁰. Another prominent framework, the Schwartz Theory of Basic Human Values, identifies ten universal values that cultures prioritize differently¹¹. While these theories provide rich, detailed maps of the moral domain, it is not clear what they imply about the appropriateness of various everyday behaviors in specific situations. Our aim is to test a parsimonious model capable of explaining broad patterns of norm variation across a large and diverse set of societies and situated behaviors¹².*"

For this purpose, we draw upon Moral Foundations Theory (MFT)¹³. MFT suggests that moral judgments are based on a set of intuitive foundations that, we argue, are also applicable to everyday behavior."

- Comment 2: Need for a primer on social norms theory.
 - Response: We have added the following sentence and citation to the first paragraph of the Introduction: *"Social norms are informal, widely shared rules that govern behavior within a group or society. Foundational work on social norms distinguishes them from formal laws, highlighting their role in maintaining social order through mechanisms of approval and disapproval.^{1"}*
- Comment 3: Circularity of the "vulgarity" argument and its relation to "inconsiderateness."
 - Response: We have rewritten this section to avoid circularity and to acknowledge the TDM perspective. The revised manuscript now states: *"Vulgarity: This refers to behaviors that are perceived as coarse, filthy, or indecent²⁰. From an MFT perspective, this concern is linked to the binding foundation of Purity²¹. Therefore, we expect it to be less impactful in societies with more individualistic morality. While we link vulgarity to the non-consequentialist concern of Purity, we acknowledge that other theories, like the Theory of Dyadic Morality, might construe such violations as a form of indirect harm (e.g., causing offense), an overlap we will return to in our discussion."*
- Comment 4: Citing emerging work on everyday moral dilemmas.
 - Response: We thank the reviewer for this helpful suggestion to better frame our contribution. We agree that citing the emerging work on everyday moral dilemmas highlights the timeliness of our research and clarifies its unique contribution. To address this, we have now added the following sentences to the Introduction, citing Yudkin et al.: *"This focus on mundane situations aligns with an emerging trend in moral psychology to move beyond classic moral dilemmas and study the more common conflicts people face in their daily lives⁹. Much of this work has focused on dilemmas within a single culture, leaving large-scale cross-cultural variation and temporal change in everyday norms under-explored—a gap the current study aims to address."*

Results

- Comment 1: Reference for different sample results.
 - Response: The Methods section now contains a paragraph called "Samples Used in the Analysis" that says: *"The analysis is performed on three samples: The Preregistered sample includes data collected until February 28 except a few societies that did not include certain questions. The All data sample also includes these societies and data collected after February 28. The Attention*

check sample excludes participants who did not pass the attention check.” We have also added an "Analysis Plan" at the start of the Results section. In that section, we note “Results reported in the text are based on the preregistered sample; using the full sample or only participants who passed an attention check yields similar results. Full results for all samples are presented in the figures and Supplementary Material.”

- Comment 2 & 3: Clarification on payment and the Ind-Bind scale's direction.
 - Response: The Methods section now starts with a subsection “Participants” clarifying that participant compensation varied by site. We have also revised the first mention of the Ind-Bind scale in the Results to note that *“higher values [indicate] more individualistic morality and lower values [indicate] greater prioritization of binding morality”*.
- Comment 4: Justification for the Ind-Bind scale's forced-choice format.
 - Response: Where the Ind-Bind scale is introduced in the Methods section, we now discuss the scale's design as follows: *“ This direct comparison format is designed to measure the relative prioritization of foundations when they conflict, a common method for assessing value hierarchies³³⁻³⁴. The scale's validity is strongly supported by its high correlation with several other related value measures collected in the study (see below and Results). ”*

Directly below, where we introduce related values measures, we highlight the alternative format of the Moral Foundations Questionnaire: *“This version included 8 items measuring the relevance of individualizing foundations and 9 items measuring the relevance of binding foundations (see the Supplementary Material), providing an indirect comparison between individualizing and binding foundations. We then subtracted the average response to binding items from the average response to the individualizing items.”*

In the “Strength and Limitations” section of the Discussion, we return to this theme: *“Finally, our study introduced a new tool, the Ind-Bind scale, to measure the relative priority societies’ place on individualizing versus binding concerns. It is distinct from the Moral Foundations Questionnaire in that it explicitly asks people to compare violations of different types of concerns instead of evaluating each concern separately. While both formats have advantages and disadvantages, we found that both scales yielded very similar results.”*

- Comment 5: Need for an analysis plan.
 - Response: We have added a new "Analysis Plan" paragraph at the beginning of the Results section to outline the structure of our analyses.
- Comment 6: Clarification on "situational dependence" variance.

- Response: We have now clarified this in the Results section: "*We defined situational dependence as the proportion of variance in a concern not explained by the behavior itself (i.e., 1 minus the variance explained by behavior in a two-way ANOVA).*"
- Comment 7: Missing error bars in Figure 4A.
 - Response: We thank the reviewer for the close reading. All estimates in Figure 4A have error bars for 95% confidence intervals, though some are very tight and therefore difficult to see. We have made the dots smaller to ensure that the error bars are visible.

Discussion

- Comment 1: Connecting unexpected findings to other moral theories.
 - Response: We have added a new section to the Discussion ("Interpretation in Broader Theoretical Context"). Here we explicitly use TDM to offer a plausible explanation for why norms against vulgarity have become stricter, linking it to an increased concern for causing offense.
- Comment 2: Missing limitations.
 - Response: We have added a "Strengths and Limitations" section to the Discussion where we now transparently address these issues, including the two-time-point analysis, the use of convenience samples, and the scope of our measured concerns.
- Comment 3: Lack of references in the discussion.
 - Response: In revising the Discussion to engage more deeply with the literature (e.g., TDM, Tightness-Looseness), we have incorporated additional citations to better connect our findings to existing work.

Methods

- Comment 1: Presenting methodological information earlier to guide readers through the results.
 - Response: As per the editor's instructions for resubmission to *Communications Psychology*, we have restructured the entire manuscript to place the Methods section before the Results section. This structural change fully addresses the reviewer's concern, as all details regarding our measures and analytical approach are now presented before the findings.
- Comment 2: Recommendation to test for and report metric and scalar invariance.
 - Response: See the new Supplementary Table 6.

Supplement

- We have added table notes and extended the x-axis of the figure as suggested. Very good suggestions!

Reviewer #2

We thank Reviewer #2 for their insightful comments and for pushing us to clarify our theoretical and methodological choices.

- Comment 1: Incomplete use of MFT. The reviewer asked why our study only seemed to pertain to 3 of the 6 moral foundations, making the dataset seem incomplete for a full test.
 - Response: Our approach is now justified in the Introduction as follows: *“The moral foundations terminology was developed for moral judgments and is therefore not directly applicable to everyday norms. Our approach is instead to identify everyday concerns that people recognize and examine whether they can be conceived as individualizing or binding concerns. We identify three primary concerns: ...”* In the Discussion we discuss the potential applicability of other moral foundations in other everyday contexts than those studied in this paper: *“The framework outlining three primary concerns about everyday behavior is novel. It may be extended to include other concerns, such as authority or loyalty, that are undoubtedly relevant in hierarchical or intergroup contexts. Such contexts were not studied here but represent an avenue for future work.”*
- Comment 2: Unclear logic for "lacking sense." The reviewer questioned the logic for why "lacking sense" would be more relevant in individualistic societies.
 - Response: We have revised the Introduction to state our logic more clearly. We now explain that the link is not direct but mediated by the "common-is-moral" heuristic: *“We hypothesize this concern will be more impactful in societies with more individualistic morality, not because of a direct link to a moral foundation, but through a stronger reliance on ‘common-is-moral’ heuristics²⁶. Prior research suggests that where individuals rely less on traditional authorities for moral guidance (a feature of individualistic morality), they are more likely to infer inappropriateness from statistical rarity or oddness²⁷⁻²⁸. A behavior that ‘lacks sense’ is likely uncommon and thus may be judged more harshly where this intuition is stronger.”*
- Comment 3: Singular focus on modernization theory. The reviewer questioned the exclusive use of modernization theory to explain change.
 - Response: This is a valid point. We have revised the lead-up to the Hypothesis on Change to acknowledge several theories of value change: *“Our framework*

predicts that everyday norms would change if the relative priority placed on individualizing versus binding concerns shifts. Several macro-level theories address such value change, proposing different drivers for this shift, such as the diffusion of global cultural scripts (World Society Theory³⁴), historical ecological pressures (Pathogen Stress Theory³⁵⁻³⁶), or socioeconomic development (Modernization Theory^{4,37-38}). While each theory offers valuable insights, it is Modernization Theory that most directly posits a continuous and directional global trend: that economic development fosters a value shift toward greater individualism, emphasizing liberty and care while the importance of tradition and purity declines. In other words, this describes a global increase in what we term individualistic morality. This clear directional prediction allows us to translate our hypothesis on societal variation into a hypothesis on temporal change.”

- Comment 4: Table 1 and levels of specificity were difficult to follow.
 - Response: We appreciate this feedback on clarity. In the revision, we have dropped the table and streamlined the text leading up to the hypothesis in order to make it more accessible to readers. The three levels of the hypothesis are in the preregistration; the hypothesis in the manuscript merges preregistered H1 (covering the overall and behavior-specific levels) and H3 (covering the situation-specific level).
- Comment 5: Systematic differences between student and non-student samples.
 - Response: In the results section for the Ind-Bind scale, we have added the following sentence: “*Differences between students and non-students were negligible (in 25 societies with both samples, Ind-Bind scores were on average 0.04 lower among students than non-students).*”
- Comment 6: Exclusion of 5 non-situational contexts. The reviewer saw this as a "forced split."
 - Response: We have elaborated on the rationale in the Methods section. We now state: “*To maintain a clear focus on situational variation, which is the core of our temporal comparison with the Gelfand et al. data, and to keep the scope of this initial report manageable, the analysis of these non-situational contexts is beyond the scope of this paper and will be addressed in a subsequent report.*” In other words, the topic of this paper is norms for situated behaviors: How appropriate is it do X on the bus/at a party/in a job interview etc.? The items we did not include in this paper are instead of the form ‘How appropriate is it do X for a woman/man’ and “How appropriate is it to do X in front of superiors/in front of peers/when no one is around”. As these manipulations do not contain a situation, they address different questions and do not fit in this paper.
- Comment 7: Necessity of reporting attention check data.

- Response: We believe this is a crucial robustness check. It is now justified in the Methods section as follows: *“The analysis is performed on three samples: The Preregistered sample (N = 17,288) includes data collected until February 28 except a few societies that did not include certain questions. The All data sample (N = 25,422) also includes these societies and data collected after February 28. The Attention check sample (N = 15,599) excludes participants who did not pass the attention check. As shown in Supplementary Table 1, the attention check pass rate varies dramatically between societies. Given this large variance, presenting the results for the attention-checked sample demonstrates that our findings are not driven by societies with lower data quality.”*
 - Comment 8: Better depiction of regional variation. The reviewer suggested a heat map for regional variation in concerns.
 - Response: It is a good idea to better visualize the regional variation in our data. The reviewer specifically suggested a heat map for the '3 categories of concerns.' However, one of our key findings is that there is extremely high cross-cultural agreement on which behaviors elicit which concerns ($r > .88$), meaning a map of perceived concerns would show very little regional variation. However, to honor the spirit of the reviewer's suggestion to geographically ground our findings, we have created a new figure (now Supplementary Figure 2) that presents a world heat map of our key predictor societal variable, Individualistic Morality (as measured by the Ind-Bind scale). This map illustrates the cultural variation that is central to our argument and provides the reader with a clear geographical context for the results that follow.
 - Comment 9: Clarification on number of ratings. The reviewer was unclear why appropriateness was rated more often than concerns.
 - Response: We have added a sentence to the Methods section to explain this design choice: *“The design choice to have fewer participants rate concerns and commonness compared to appropriateness was made to minimize participant fatigue while still gathering robust data on our primary dependent variable (appropriateness) and the characteristics of the stimuli (concerns).”*
-

Reviewer #3

- Comment: The reviewer co-reviewed the manuscript as part of a training initiative.
 - Response: We thank Reviewer #3 for their time and contribution to the review process.
-

Reviewer #4

We thank Reviewer #4 for their comments on the paper's theoretical framing and contribution.

- Comment 1: Paper feels descriptive; main takeaway is unclear.
 - Response: Our theoretical goal was to show that variations and changes in norms are not random but are systematically structured by underlying moral values and everyday concerns. The fact that we can explain a significant portion of this variation with a parsimonious model is, we believe, a significant explanatory contribution. We have revised the Introduction and Discussion to make this explanatory, rather than descriptive, contribution clearer.
- Comment 2: Unclear MFT concepts and lack of engagement with alternative frameworks (TDM, Tightness-Looseness).
 - Response: This is a crucial point that overlaps with feedback from Reviewer #1. Please see our detailed response to Reviewer #1, Comment 1 and Reviewer #1, Discussion Comment 1. We have revised the manuscript to broaden the theoretical context.
- Comment 3: Confusing metaphors and jargon.
 - Response: We thank the reviewer for this feedback. In the revision, we have removed the "moral taste buds" metaphor and have worked to reduce jargon throughout the manuscript to improve accessibility.
- Comment 4: Strong, under-justified assumption about cross-cultural agreement on concerns.
 - Response: We agree this assumption requires empirical support. We now state in the Introduction that it is an assumption we explicitly test. In the Results, we present the strong evidence that validates it: *"We found extremely high cross-cultural agreement on which concerns are elicited by which behaviors. Correlations between societies with high vs. low individualistic morality were $r=0.96$ for vulgarity, $r=0.93$ for inconsiderateness, and $r=0.88$ for lacking sense, validating our assumption that concerns are perceived similarly across cultures."*
- Comment 5: Need for more controls (political ideology, religiosity).
 - Response: Political ideology and religiosity are not conceptually distinct from individualistic morality and therefore not suitable as controls. Indeed, we use religiosity to establish convergent validity of our measure of individualistic morality. Note that a supplementary analysis (Supplementary Figure 4) shows results are robust to controlling for society-level GDP and a full set of individual-level demographic variables.

- Comment 6: Unclear theoretical motivation and competing hypotheses.
 - Response: As the title of our paper says, this project addresses - both theoretically and empirically - how everyday norms vary across behaviors, situations, societies, and time. We have revised the Introduction and Discussion to provide clearer theoretical motivation and to engage with alternative theoretical perspectives, thereby clarifying the scholarly landscape and the contribution of our specific test.

RESPONSE TO REVIEWERS

Reviewer #2 (Remarks to the Author):

I think that the authors have done a fine job addressing my comments in the revision.

AUTHORS: Thank you!

Reviewer #3 (Remarks to the Author):

I really commend the authors for their thoughtful and thorough revisions. They responded carefully to each of my concerns, and I feel the paper is now much better situated within the broader context of research on moral psychology beyond MFT alone. It's clear they put great care into strengthening the manuscript, and I believe it will make a wonderful contribution to the field.

AUTHORS: Thank you!